# Review on Preformed Crowns in Pediatric Dentistry—The Composition and Application

**DOI:** 10.3390/ma15062081

**Published:** 2022-03-11

**Authors:** Klaudia Sztyler, Rafal J. Wiglusz, Maciej Dobrzynski

**Affiliations:** 1Department of Pediatric Dentistry and Preclinical Dentistry, Wroclaw Medical University, Krakowska 26, 50-425 Wroclaw, Poland; 2Institute of Low Temperature and Structure Research, Polish Academy of Sciences, Okolna 2, 50-422 Wroclaw, Poland

**Keywords:** pediatric crowns, primary teeth, material composition, teeth restoration, repair procedures

## Abstract

The purpose of this review is to compare and contrast the various types of preformed crowns that can be used to restore the primary teeth in children. Historically, preformed crowns have been widely available for the past 50 years. The clinical performance of preformed crowns has evolved to meet higher functional, mechanical, and aesthetic demands. Preformed crowns are available in a range of prefabricated sizes and shapes. Preformed crowns can vary depending on their properties, compounds, methods of preparation, and biocompatibility.

## 1. Introduction

Dental caries is one of the most widespread medical conditions both in adults and children [1]. According to WHO Oral Health facts, more than 530 million children suffer from dental caries of primary teeth [2]. A survey from 2019 conducted by Public Health England shows that one in four five-year-olds have had dental caries [3]. The National Dental Inspection Program in Scotland found that 15% of Scottish children had at least one tooth extracted due to caries before the age of five [4]. This number increases to 42% for eight-year-olds [5]. The consequences of primary tooth decay include local and systemic problems. Local potential problems include pain due to pulp or periapical tissue inflammation and infection of permanent tooth buds, which can interfere with odontogenesis and cause a defect called Turner’s tooth. Premature loss of primary teeth can potentially cause malocclusion, tongue movement disorders, chewing disorders, change of facial features, and behavioral difficulties. The presence of caries in primary teeth increases the risk of caries in permanent teeth. Systemic implications include symptoms of infection such as high temperature and apathy. Long-term pain can lead to eating difficulties, which can cause weight loss and growth and development disorders. This illustrates how crucial it is to keep primary teeth healthy and prevent the development of caries [6]. If caries develops, it is highly recommended to implement treatment as soon as possible. The treatment options for irreversible caries contain nonrestorative cavity control [7] including the Hall technique [8] and techniques associated with the removal of caries. Removal can be performed selectively, either through selective caries removal or stepwise caries removal [9,10], or non-selectively by removing all demineralized dentin. The use of the latter-mentioned procedure has not been recommended [11]. After preparation, the dental tissue must be restored. Choosing the best method of restoration is another important step to provide the best treatment results. The most chosen restoration material is glass ionomers, composite resin, compomer, and amalgam, or prefabricated crowns [9]. Many studies compare these materials to each other in terms of durability, secondary caries, endodontic complications, or restoration loss. According to many studies, the best outcome is achieved by using prefabricated crowns [12,13,14]. A study by Isabel Cristina et al. analyzed the survival rate of stainless-steel crowns compared to composite restorations in primary teeth undergoing endodontic treatment. The study showed that the overall success rate was higher in the SSC group (88%) compared to composite restorations (75%), but more importantly, failure of composite restoration resulted in endodontic treatment failure, whereas endodontic failure in SSCs group was not associated with restoration failure [15]. Stainless steel crowns present the highest success rate, with a performance level of 96.1%. In comparison, resin modified glass-ionomer shows a 93.6% success rate, compomer 91.2%, and metal-reinforced glass ionomer cement 57.4% [16]. Roberts et al. compared the survival of resin-modified glass ionomer cement (RMGIC) to PMC. Their study shows that PMCs present a very high survival rate, especially for large cavities and teeth that have undergone endodontic treatment [13]. It is also worth noting that prefabricated crowns can be used to restore non-carious lesions or developmental defects such as hypoplasia, hypomineralization, etc. [17]. 

## 2. Materials and Methods

Pediatric crowns are a prefabricated solution for full crown restoration of deciduous teeth. They are available in sets containing different sizes and shapes dedicated to primary teeth. The first commercially available primary crowns were made of steel and contained large amounts of nickel. Subsequent generations of crowns had improved the composition of metal crown material, as well as enhancement of their prefabricated shape. Today, most pediatric steel crowns are made of stainless steel, however, their compositions can vary. Due to the poor aesthetics of grey stainless-steel crowns, patients prefer white crowns to match the color of other natural teeth. Most crowns include pre-veneered stainless-steel crowns (PVSSCs), crowns made of polymers, pre-veneered aluminum crowns, or prefabricated zirconia crowns. These types of crowns are intended to mimic the natural color of the teeth, which is highly desired by parents, especially in the anterior segment of the dental arch. The use of crowns allows the reconstruction of teeth with severe damage, both caused by caries and processes associated with the disruption of hard tissue development. They are often the best, or one of the few solutions determining clinical success and maintenance of the tooth in the oral cavity until physiological tooth replacement.

This review will focus on basic types of prefabricated pediatric crowns by comparing their characteristics, indications, contraindications, advantages, and disadvantages.

It should be noted that the quantitative compositions of individual brands of preformed pediatric crowns can vary. The information on composition was gathered from product safety data sheets. The selected publications contain a thorough description of each group of crowns, their properties, advantages, disadvantages, indications, contraindications, preparation protocols, and treatment results (Figure 1) [18].

## 3. Crowns

Prefabricated crowns have been widely used in pediatric dentistry for the last 50 years [19]. The three recently most used ones are preformed metal crowns, resin veneered stainless steel crowns, and strip crowns [20]. More aesthetic solutions include pre-veneered crowns and zirconia crowns. The use of pediatric crowns involves the proper preparation of the tooth crown for it to fit well, or it may involve no preparation and the use of the Hall technique. In the Hall technique, the reconstruction of the tooth is performed without local anesthesia by placing the crown on the remaining tooth tissue and pressing it to the correct position using finger pressure or the patient’s occlusion force. The use of crowns is especially recommended for teeth after pulp treatment or with advanced decay damage. They are also a good solution in the case of developmental disorders of dental hard tissue. They can also be used as a method of reconstruction of deciduous teeth during procedures under general anesthesia. Their primary purpose is to allow tight restoration, with a long-term positive outcome and without major failures. The overall procedure should cause as little pain and as little trauma for the young patient as possible. Innes et al. in their systematic review conclude that the use of crowns is associated with a reduced risk of major failure, pain, and formation of abscess in the long term compared to conventional restorations. The use of crowns may be associated with a higher risk of gingival bleeding [17]. There are a small number of studies comparing different types of crowns. Therefore, we cannot, in the current state of knowledge, say which crowns are best. This refers to the use of zirconia crowns as a replacement for stainless steel crowns [21]. Pediatric crowns should be easy to adapt, the bonding strength should be high enough to withstand masticatory forces. Crown materials should be safe for antagonist teeth. They should not hamper oral hygiene maintenance and should be biocompatible with the surrounding tissue [22,23,24]. Sahana S et al. divide crowns into two groups (see Figure 2) [25].

### 3.1. Preformed Metal Crowns

Preformed metal crowns (PMCs) can be divided into two groups depending on the composition. The first group is represented by stainless steel crowns (SSCs), while the second group include nickel-chromium crowns [19]. SSCs were introduced to dentistry in 1950. Their first prototypes were usually too large, with straight and very long sides. Proper adaptation to teeth required many steps such as trimming, contouring, crimping, and finishing [26,27]. The next generation of SCCs was focused on improving the imitation of natural tooth anatomy, which would help simplify the adaptation process. Introducing nickel-chromium crowns helped eliminate many disadvantages found in the first SSCs. Firstly, Cr-Ni crowns are fully shaped and resistant to defects [27]. Secondly, thanks to improved anatomical accuracy, they rarely require trimming [27]. It is worth noting that they also need to be modified to improve adaptation, but they usually need fewer steps to achieve it. Figure 3 presents an example of modern preformed metal crowns available on the market (below). 

Preformed metal crowns (PMCs) are represented on the market by three main types of crowns. The first is untrimmed crowns. They require long adaptation due to a lack of trimming and contouring in the production process. An example of these crowns is Rocky Mountain Crowns. The second type are pre-trimmed crowns. The sides of these crowns are straight and festooned to follow the gingival crest line. They still require contouring and sometimes need to be trimmed [28]. The third type are pre-contoured crowns. The sides of these crowns are pre-contoured and festooned. They show the best imitation of anatomical geometry, yet occasionally minimal trimming and recontouring are required in the adaptation process. An example of these crowns is Unitek Stainless Steel Primary Crowns by 3M [28]. PMC usually contains 67% iron, 10–13% nickel, 17–19% chromium, and 4% of minor elements, although each brand available on the market has a slightly different composition in Table 1.

Indications for use in primary molar teeth [19,27,28,34]:Decay affecting two or more tooth surfaces;Inability to place an amalgam filling;Restoration after pulp treatment procedures;Restoration in non-carious lesions or developmental defects;Restoration of fractured primary molars;Severe bruxism;Restoration in children who require general anesthesia for treatment;In children with a high risk and high susceptibility to caries;An abutment for a space maintainer.

Indications for use in permanent molar teeth [19]:Temporary restoration after tooth fracture;Temporary restoration until orthodontic opinion and treatment plan;Temporary restoration before final prosthetic restoration;Economic reasons;Restoration in non-carious lesions or developmental defects;Restoration of a permanent molar that must cover the entire crown.

Contraindications for the use of PMC [28]:Allergy or vulnerability to nickel;Uncooperative patient;A primary tooth near its exfoliation time;A radiograph showing resorption of more than half of the tooth root.

The main advantages of PMCs are their capability of tight and long-term survival restoration even when the risk of caries is high. The application procedure is simple, which lowers the possibility of errors during the treatment. Lastly, they have the advantage of multiple clinical indications [13,14,19,35]. Their main disadvantage is low aesthetics. They also cannot be used on partially erupted teeth [19]. Possible complications during the preparation and application of PMCs are crown draft, usually towards a massively destructed wall, interproximal ledge, or poor margins when crowns are incorrectly adapted [36,37,38,39,40].

Preparation [27,28] (Figure 4):Prior to placing the PMC, the dentist should discuss the treatment with the child and the parents/guardians and obtain their consent;The dentist should estimate the crown size, to enable it to click into place. When choosing the appropriate crown size, it is recommended to measure the mesial-distal width between the contact points of the adjacent teeth with calipers. If this cannot be achieved, the mesial width of the contralateral tooth in the opposite arch can be measured. It is advised that the smallest matching crown should be selected;Local anesthesia and tooth isolation;If necessary, caries removal, pulp therapy;Tooth restoration with glass-ionomer cement or compomer;Occlusal reduction of about 1.5 mm;Mesial and distal reduction, so that the probe can pass through, maximally 1 mm;No buccal and lingual reduction or minimal reduction;Try in of the crown; the crown should go maximally 1 mm subgingival, if it goes deeper, it requires adaptation. Trimming is performed with special crown scissors or an abrasive wheel. After trimming, the crown needs to be crimped with crimping pliers. Finally, the margins should be thinned with white stone and finely polished;Cementation with the use of resin-modified glass ionomer, polycarboxylate phosphate cements, or RelyX™ Luting Plus Cement. Placing is usually performed from the lingual side and rolled during the preparation to the buccal margin.

**Figure 4 materials-15-02081-f004:**
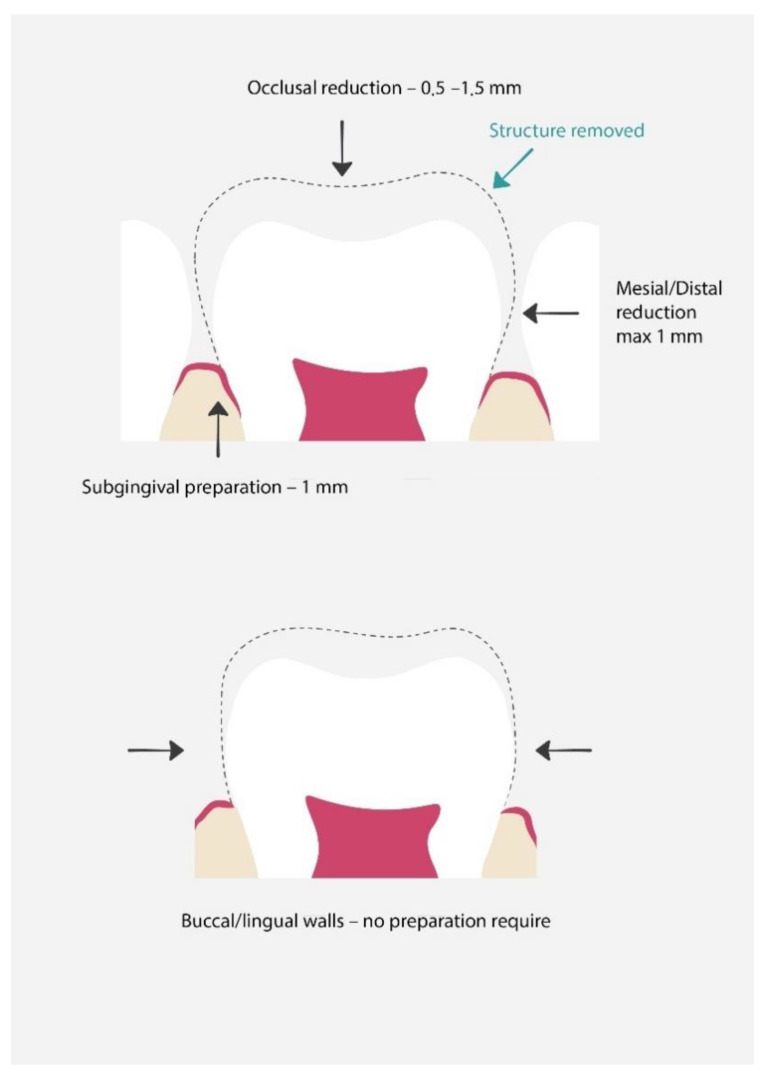
Proper preparation for preformed metal crowns. The dashed line shows the range of hard tissue removal, which will later be used for proper crown fit and restoration [25].

Sometimes crown adaptation involves various difficulties. If the smallest size of PMC is too large to be used for restoration, metal edges can be sized down by cutting and overlapping, which results in the reduction in crown circumference. In the next step, overlapped margins are welded together [27]. On the other hand, if the largest PMC available is too small, the best fitting crown can be cut, and filled with an additional piece of orthodontic stainless steel band material welded over the space [41]. When multiple crowns are fitted and adjusted, it is recommended to reduce proximal surfaces more than usual [42]. Another option for adjusting PMC is the Hall technique.

### 3.2. Hall Technique

The Hall Technique is a method of adjusting and establishing PMCs which belongs to nonrestorative cavity control techniques [43]. The first step of this technique is to separate the tooth by using dental floss or separating pliers [44,45]. Separators are left in for approximately five days. After that, the separators are removed, and the proper size of the crown is estimated and placed on a tooth with glass ionomer cement [8]. The crown is fitted on a tooth with the pressure exerted by the dentist with his or her fingers or with the child’s bite force [46]. The last step is the removal of excess cement. This procedure does not require local anesthesia, caries removal, or tooth preparation. The idea of this technique is to arrest caries and force them to change into less cariogenic flora, which will stop or at least slow down the progression of caries [47]. 

Indications for the use of the Hall technique in primary molars [44]:Used in occlusal caries or non-cavitated teeth, if the patient is unable to tolerate fissure sealant, partial caries removal, or conventional restoration;Proximal caries or non-cavitated teeth if the patient is unable to tolerate partial caries removal or conventional restoration.

Contradictions for the use of the Hall technique in primary molars [7]:Pulp infection;Irreversible pulpitis;Pulp exposure;Lack of clear band of dentine on the radiograph;Clinical or radiological signs of peri-radicular pathology;Extremely damaged crowns.

This technique presents many advantages. Firstly, it is very non-invasive, because it does not require the injection of local anesthesia, caries removal, or tooth preparation [48]. The procedure is quick and less traumatic, which can improve the child’s future cooperation [48]. Some authors show that it is also more cost-effective than conventional restorations [49,50]. On the other hand, this technique is controversial and causes some concerns. Firstly, there is no tooth preparation which leaves the crown without additional space, which in turn leads to premature occlusal contact after cementation. In the days following the procedure, biting forces resolve premature contact and usually after one or two days the juvenile patient does not feel discomfort [43]. It is worth noting that further studies are needed to analyze the effect of the Hall technique on occlusion and the temporomandibular joint [46]. Another important disadvantage is the use of non-aesthetic PMCs. Overall, the Hall technique shows promising results. Data show that it can provide a lack of pain and infection and general high effectiveness [51,52,53].

### 3.3. Open-Faced Stainless Steel Crowns

This is a form of the use of SSCs in the anterior section of the dental arch. The procedure includes adapting proper SSC. If needed, the crown is trimmed, crimped, and polished. After the crown is cemented and the cement sets, the labial wall of the crown is cut out and the luting cement is partially removed to create undercuts. In the following step, the space is filled with a more aesthetic material such as composite [19,54].

Indications to use open-faced SSC [54]:Crown fracture;Pulp protection.

Contraindications to use open-faced SSC [54]:Allergy or vulnerability to nickel;Uncooperative patient;A primary tooth near its exfoliation time;A radiograph showing resorption of more than half of the tooth root;Tooth fracture level below gingival margin.

Their main advantage is better aesthetics compared to traditional SSCs, however, the procedure is time-consuming and requires a dry restoration area. The restoration may have poor color stability and the metal margins of the crown might still be visible [55].

### 3.4. Pre-Veneered Stainless Steel Crowns

PVSSCs combine the mechanical properties of SSCs with the additional aesthetic factor of composite resin or thermoplastic resin [19]. The aesthetic part is either chemically or mechanically bonded to the crown [56]. At first, the restoration of anterior primary teeth were introduced to their indications; later on, they were also developed to restore primary molars [19,54,56]. The examples of PVSSCs available on the market are Nusmile Primary Crowns, Kinder Krowns, Cheng Crowns, Flex Crowns, Dura Crowns, and Whiter Biter [56,57]. The exemplary composition of these crowns presented in Nu Smile Pediatric Crowns safety data sheet contains composite paste, iron, copper, silver, 2-hydroxyethyl methacrylate, chromium, nickel, zinc, manganese, silicon, molybdenum, cobalt, and carbon [58]. A study by Sean Beattie et al. compared three pre-veneered stainless steel crown manufacturers for their fracture resistance. Their study involved EC crowns, Kinder Krowns, and NuSmile Primary Crowns. The crowns were subjected to uniaxial force. The results showed no significant differences in fracture resistance between the crowns tested, and furthermore, the forces required for fracture in each case exceeded the control child’s occlusal force in the 6- to 10-year age range [59]. Their advantages are long durability and a good aesthetic. PVSSCs allow restoration when the treatment area cannot be perfectly dry [19,60]. On the other hand, they require more aggressive tooth preparation compared to SSCs. They come with some limitations such as prefabricated resin shade, which can look artificial [19]. They are also wide mesio-distally, which can cause problems with placing them in patients with crowding [19,54]. The labial section cannot be crimped, because it might weaken the aesthetic facing and cause premature failure [19,56]. It is also worth noting that clinically try-in crowns that do not meet the proper parameters and require sterilization procedure, which can exert stress on the resin [56]. To reduce the impact of stress, it is recommended to use steam sterilization [56].

Preparation [56] (Figure 5):Discuss the procedure with the parents and child and obtain their consent;The dentist should estimate the crown size;Local anesthesia and tooth isolation;Occlusal reduction around 2 mm or incisal reduction around 2 mm;Circumferential reduction 25–30%;In posterior teeth buccal reduction 1.5–2 mm;Feather-edge subgingival preparation 1.5–2 mm;If necessary, removal of caries and pulp therapy;Try in of the crown;Cementation of the crown. The cement of choice is glass-ionomer.

**Figure 5 materials-15-02081-f005:**
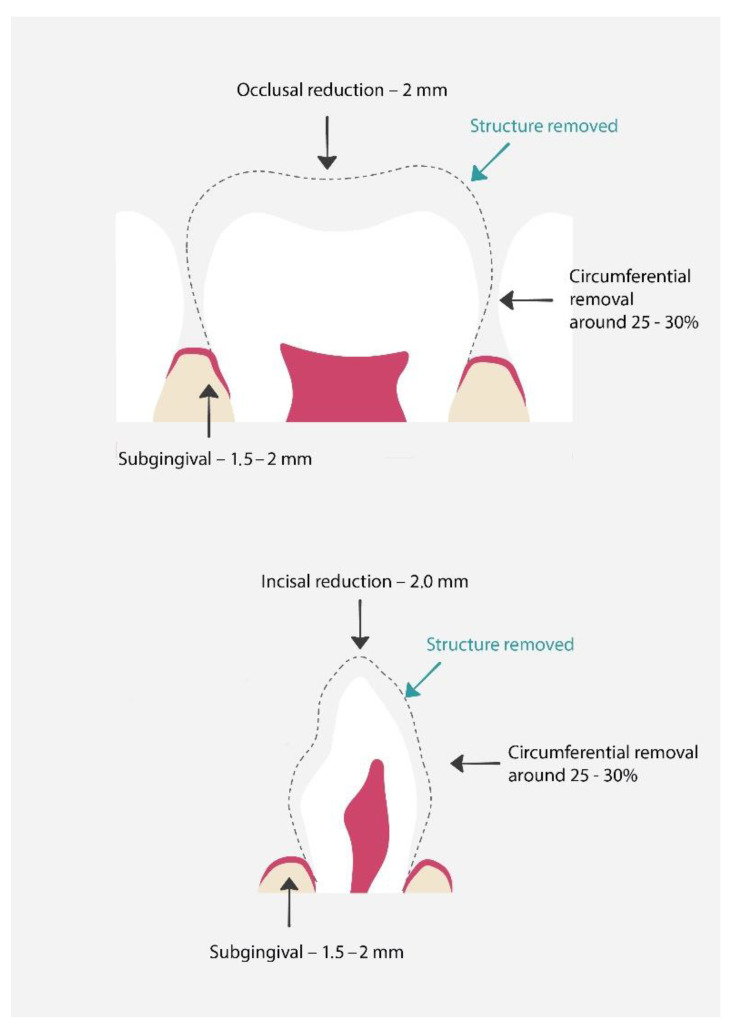
Proper preparation for pre-veneered stainless-steel crowns. The dashed line shows the range of hard tissue removal, which will later be used for proper crown fit and restoration [61].

### 3.5. Pedo Pearl

These crowns can be included in the group of pre-veneered crowns. The base of these crowns is made of aluminum covered with epoxy paint which gives them an aesthetic tooth color [54]. They are easy to adapt by cutting and crimping [43]. If necessary, they can be covered with composite [62]. Their disadvantages are their soft structure and possible shorter durability [63].

### 3.6. Polycarbonate Crowns

Anterior primary teeth are usually damaged due to early childhood caries (ECC) caused by bottle feeding without proper hygiene. ECC usually starts on the labial surface of the upper incisors and progresses rapidly. The treatment depends mostly on the cooperation with the patient and includes non-restorative cavity control, preparation, and restoration with conventional materials. Crowns can also be used for restoration and are especially useful when the caries damage is extensive and conventional restoration might be problematic. Polycarbonate crowns are made of aromatic polyesters of carbonic acids [19,54]. They can be described as thermoplastic resins. The use of high temperature (around 130 °C) and pressure makes the material easy to mold and shape into the desired form [19,54]. The material properties are thin structure and flexibility greater than that of acrylic resin crowns [64]. On the other hand, these crowns do not resist high abrasive forces which can cause fracture or premature crown loss [64]. Figure 6 presents an example of polycarbonate crowns available on the market (below).

Indications for the use polycarbonate crowns [19,54]:Full restoration of anterior teeth destroyed by caries;ECC as lesion stabilization;Discolored teeth;Restoration after pulp therapy;Restoration in non-carious lesions or developmental defects;Abutment for space maintainers.

Contraindications to use polycarbonate crowns [54]:Too small teeth;Crowded anterior teeth;Excessive tooth damage preventing retention;Bruxism;Excessive abrasion;Overbite;Deep impinging bite.

Preparation [61] (Figure 7):Discuss the procedure with the parents and child and obtain their consent;The dentist should estimate the crown size; the most important part is to properly estimate the mesiodistal dimensions to obtain proper tooth contour;Local anesthesia and tooth isolation;Incisal reduction by about 2 mm;Tiny mesiodistal preparation. The walls should be slightly parallel;Facial/Lingual reduction by about 1 mm;Old protocols suggest performing a chamber 1 mm below gingiva on labial and proximal surfaces;Feather subgingival preparation 1 mm;If pulp procedures had been performed on the tooth, the lingual opening can be used as additional retention;Crown fitting;Cementation;Removal of excess resin cement.

**Figure 7 materials-15-02081-f007:**
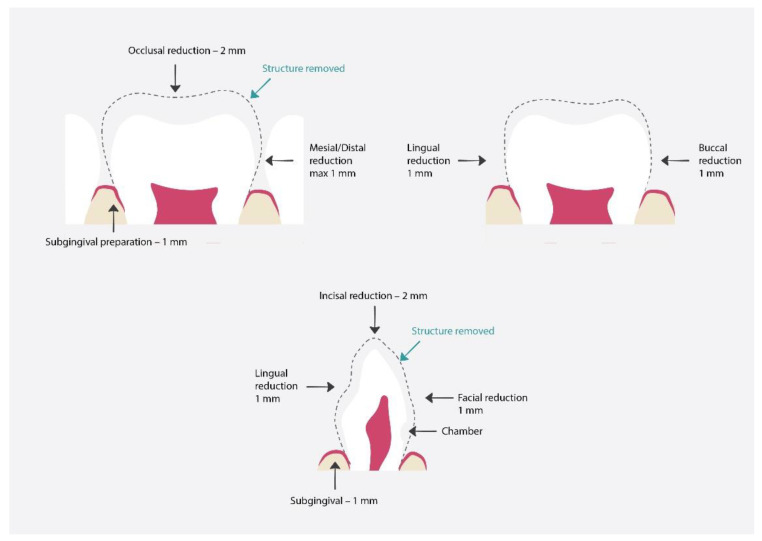
Proper preparation for polycarbonate crowns. The dashed line shows the hard tissue removal range, which will later be used for proper crown fit and restoration [64].

Polycarbonate crowns are at risk of fracture, dislodgement, and discoloration, all of which contribute to a possible unsuccessful treatment outcome [64]. For this reason, many clinicians choose to use a different type of polymer crown type, strip crowns [61]. 

### 3.7. Strip Crowns

Strip crowns are transparent plastic forms used to simplify work within upper incisors restoration. They can be filled with both chemical and light curing composite material. Once the material has set, they can be easily removed, leaving a smooth surface. According to Kupietzky et al., their advantages ease of fitting, trimming, and removal [65]. They are also thin and transparent, which makes them easier to match to natural dentition and control composite color. For the best treatment results, patients require proper hygiene instructions and further proper hygienization. Parents must be aware that the lack of proper oral hygiene decreases the chances of successful treatment, which means that they are partially responsible for the procedure’s overall outcome [66]. The surrounding soft tissue must be free of inflammation [66]. Strip crowns provide high aesthetics and functionality. They are also cheap and easy to repair [65]. However, their disadvantages include the need to maintain a dry restoration area. Any moisture or blood can interfere with the bonding, and blood can also cause discoloration of the composite material [67]. Their use is also restricted to primary teeth having enough enamel to allow proper bonding after preparation [54]. Minimal reduction is required for proper preparation [61].

Preparation [54]:Discuss the procedure with the parents and child and obtain their consent;Local anaesthesia and tooth isolation;The dentist should estimate the crown size. To facilitate crown size selection, the length of the incisal edge of the tooth being treated or—if the tooth is damaged—of the matching tooth can be used;Reduction in tooth length;Mesial-distal preparation;Knife edge preparation at gingival margin;Choosing composite shade;Preparing vent holes in incisal corners;Firmly seating the crown with composite on the tooth;Curing the composite;Strip crown removal. For safety, the best way is to use a hand piece such as a carver.

Indications to use strip crowns [66]:Extensive decay of the primary anterior teeth;Fractured teeth;Restoration in non-carious lesions or developmental defects;Teeth discoloration;Teeth after pulp therapy.

Contraindications to use strip crowns [66]:Significant teeth tissue loss preventing proper retention;Deep overbite;Periodontal disease.

### 3.8. Pedo Jacket Crown

Similar to strip crowns [19], Pedo Jacket Crowns primarily differ in the material used. They are made from tooth-colored copolyester and filled with resin material. Pedo Jacket Crowns are only available on the market in a single-color shade. Another difference compared to strip crowns is that Pedo Jacket crowns are left on the tooth after polymerization [62]. They cannot be adapted by trimming and reshaping with high-speed finishing bur, as doing so would melt the copolyester [50].

### 3.9. New Millennium Crowns

The New Millennium Crowns are made of laboratory-improved composite resin material [19,54] and are also similar to strip crowns. Their advantages include high aesthetics and parental satisfaction [68]. They can be adapted by reshaping them with a high-speed bur, however their disadvantages include a fragile structure, the need for a dry restoration area, as well as the possible discoloration of the crown by the hemorrhage [19]. They also cannot be crimped [62]. Preparation for New Millennium crowns is similar to strip crown preparation [54].

Indications [19]:Restoration of multi-surface caries;Discolored primary incisors;Anterior teeth fracture;Restoration in non-carious lesions or developmental defects.

Contraindications [19]:Difficulty in keeping the restoration area dry;Overbite;Deep impinging bite;Extensive tooth damage that prevents retention;Periodontal disease.

### 3.10. Artglass Crowns

Artglass Crowns, also known as Glastech, are made up of polymer glass which forms a three-dimensional molecular network with a cross-linked structure [54]. They contain such fillers as micro-glass and silica, which improve their durability and aesthetics compared to strip crowns [54]. Their longevity is comparable to that of porcelains [62].

### 3.11. Zirconia Pediatric Crowns

Zirconia has three forms including a monoclinic, tetragonal, and cubic one [69]. These structures are stable in various temperature ranges. The Monoclinic form is stable at room temperature; above 1170 °C, zirconia changes into a tetragonal form, while at 2370 °C, the main form is the cubic one [70]. When zirconia is cooled, the tetragonal phase changes into the monoclinic phase, causing a volumetric expansion of 3–4% [70]. In dentistry, zirconia is used in the form of yttria-stabilized tetragonal polycrystal (Y-TZP), magnesia-partially stabilized zirconia, and zirconia-toughened alumina [69]. Zirconia has many beneficial properties. Firstly, it is very strong, and secondly, it offers good aesthetic properties and good biocompatibility. Zirconia shows high wear and corrosion resistance [69]. It can also resist crack propagation due to a change in the crystalline phase [71]. Zirconia pediatric crowns require minimal preparation; moreover, the whole preparation and restoration process can be completed during a single visit. They are also an alternative for patients with Ni-Cr allergy or sensitivity. Their disadvantage is their high cost. While they cannot be modified, they also show greater thickness than PMCs [70]. According to Sumer et al., zirconia crowns exhibit less plaque accumulation, as evidenced by follow-up visits. They also show nearly zero risk of developing secondary caries and significantly lower restoration loss rate in comparison to strip crowns [71]. A study by Pinar et al. shows that plaque index and gingival index exhibit lower values around zirconia crowns compared to SSCs [72]. This results in better gingival health [73]. Zirconia pediatric crown brands available on the market include Ez-Pedo, NuSmile ZR, and Kinder Crowns Zirconia (see Table 2).

Indications for the use of zirconia crowns [19]:Decay affecting two or more teeth surfaces;Inability to use amalgam restoration;Restoration after pulp treatment procedures;Restoration in non-carious lesions or development defects;Restoration of fractured primary molars;Restoration of fractured anterior teeth;Bruxism;Restoration in children who require general anesthesia treatment;In children with high caries risk and tendency;An abutment for a space maintainer;Discolored primary incisors.

Preparation [69] (Figure 8):Discuss the procedure with the parents and child and obtain their consent;Local anesthesia and tooth isolation;Reduction in incisal wall of around 1.5–2 mm or occlusal reduction around 2 mm;Buccal reduction around 0.5–1 mm, lingual reduction around 0.75–1.25 mm;Knife edge subgingival preparation 1–2 mm;Checking the occlusion to see if there is adequate clearance from the opposing dentition;Cron selection. This can be achieved by placing the incisal edge of the zirconia crown against the incisal edge of the identical tooth;Cementation with the use of resin-modified glass-ionomer or calcium aluminate cement;Removal of excess cement.

### 3.12. Summary of Crowns Used in Pediatric Dentistry

The following Table 3 provides summary of crowns used in pediatric dentistry presenting their advantages and disadvantages.

## 4. Risks of Using Pediatric Preformed Crowns

### 4.1. Periodontal Aspects

Many researchers present a link between restoration using pediatric crowns and plausible periodontal complications. It is worth emphasizing that patients in need of pediatric crowns might lack an oral hygiene routine, which might increase both plaque accumulation and caries risk. This aspect prompts us to properly educate both patients and their parents. Preventive hygiene instructions should be included as a first step in the treatment plan [77,78]. The second aspect that might increase gingivitis risk around pediatric crowns is the inadequate contour of the crown margins [79,80,81,82,83]. Goto found out that posterior crowns presented a higher percentage of gingivitis, which could be caused by more difficult access and the fit of the crown itself [84]. Many studies show that good or moderate fitting of the crown does not significantly increase gingival problems or plaque accumulation [76,77,78,79,80]. The last factor that can be associated with gingivitis is the presence of residual cement left in the gingival pocket [85,86].

### 4.2. Nickel Allergy and Sensitivity

Nickel percentage content in PMCs changed over the years. Modern PMCs contain around 5–12% nickel, significantly less compared to old formulation nickel-chromium crowns. A study by Feasby et al. shows that a group of children who received old formulation crowns presented an increased nickel-positive patch test, whereas children with modern PMCs showed no statistical difference compared to a control group with no history of nickel appliance use [87]. Nickel hypersensitivity is more frequent in females than males. This relates to ear piercing and the usage of jewelry containing nickel. A study by Keruso et al. and Hoogstraten et al. reported that orthodontic treatment with nickel-containing stainless-steel appliances before ear piercing shows a lower risk of nickel hypersensitivity [88,89]. It is also worth noting that any adjustment to a crown, including cutting or crimping, might increase the risk of corrosion, so margins should be smoothed and polished to a high gloss to minimize this process [90,91,92]. According to Leila Basir et al., the number of released nickel ions decreased with the trimming of margins. They also noticed increased nickel release as the temperature increased [93]. A study by Dr. Deepak Bhayya et al. shows a difference in nickel ion release from 3M ESPE stainless steel crowns depending on the pH. The results show significant nickel ion release of pH 4.3, 5.5 and 6.3 in artificial saliva, with a maximum nickel release of pH 4.3, followed by 5.5 and 6.3 [94]. According to a survey by the Clinicians Report Foundation, allergy to SSCs is very rare [61].

### 4.3. Biological Response

Every material introduced to the oral environment can be associated with a biological response. PMCs contain various heavy metals, which might be released due to mechanical, chemical, and thermal intraoral stimulation. While experiments performed on laboratory rats showed that metal pieces containing Fe, Cr, and Ni were found to be cytotoxic to DNA and cultured cells [95], the amounts of metal ions present were not harmful to human health [96,97]. Various human studies show an increase in Ni, Cr, and Fe ions in the saliva; however, although the maximum amounts were always lower than dietary intake and were not capable of causing toxicity, further studies are needed [98,99,100,101,102,103,104,105,106].

## 5. Conclusions

In pediatric dentistry, a great deal of effort is required to obtain proper restoration. The first difficulty is to ensure the child’s cooperation due to their age; a fear of dentistry is a frequent obstacle in the treatment process. Therefore, it is vital for treatment procedures performed in pediatric dentistry to utilize the simplest and least traumatic procedures that have the best prognosis regarding long-term durability. This review presents one of the approaches for full coverage restoration in pediatric dentistry practice using different types of preformed pediatric crowns. The use of pediatric crowns makes it possible to achieve long-term and positive therapeutic effects. Their use is often easier and faster than manual reconstruction, especially in the case of multi-surface cavities. They are also a good option for restoring teeth after pulp treatment and those with abnormal hard tissue development. PMCs are the most used crowns. Currently, these crowns contain very low levels of nickel and are associated with a low risk of nickel allergy and hypersensitivity. It should be mentioned that there are many types of PMCs available on the market. They differ in terms of composition and prefabrication methods. Therefore, it is important to select solutions corresponding to the dental operator’s needs and demands. Over the years, new types of pediatric crowns were introduced to the market to overcome the disadvantages of stainless-steel crowns and respond to the demand for higher aesthetics, such as zirconia crowns. An important factor that requires further investigation is a comparative analysis of the use of crowns from different materials, such as a comparison of the long-term durability of zirconia crowns in comparison with PMCs. Ultimately, it is always up to the dentist and the parents to choose the optimal type of restoration based on the child’s cooperation, as well as the parents’ aesthetic requirements and economic factors.

## Figures and Tables

**Figure 1 materials-15-02081-f001:**
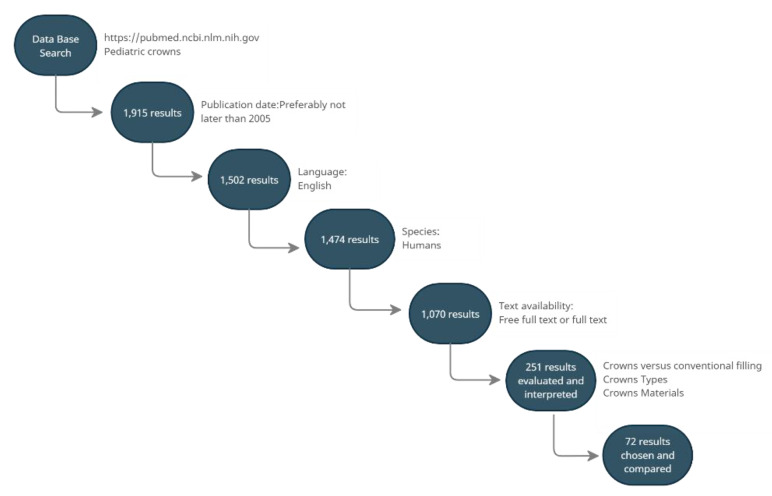
A flowchart of the database search strategy (accessed on 1 August 2021).

**Figure 2 materials-15-02081-f002:**
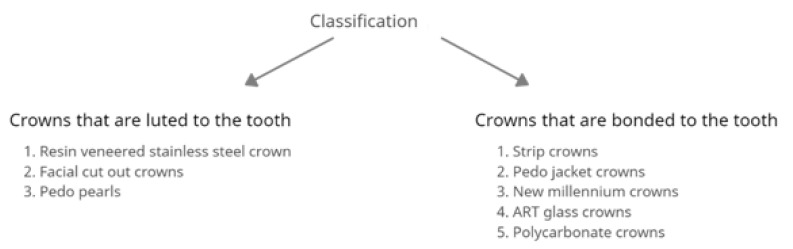
Classification of crowns used in pediatric dentistry.

**Figure 3 materials-15-02081-f003:**
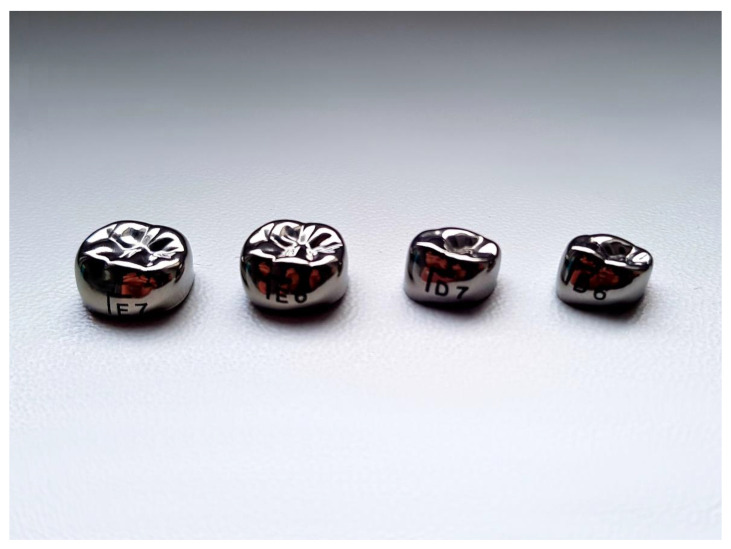
Preformed metal crowns.

**Figure 6 materials-15-02081-f006:**
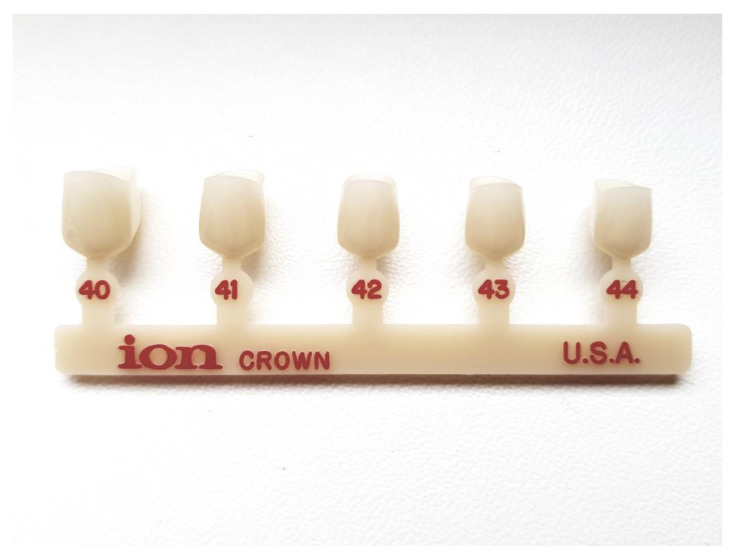
Polycarbonate crowns.

**Figure 8 materials-15-02081-f008:**
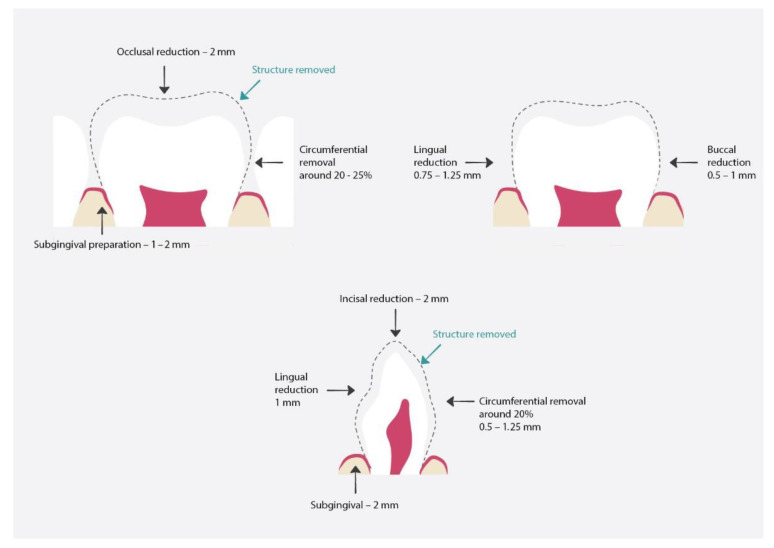
Proper preparation for zirconia crowns. The dashed line shows the hard tissue removal range, which will later be used for proper crown fit and restoration [69].

**Table 1 materials-15-02081-t001:** The composition of preformed metal crowns.

Brand	Composition	Toxicity	Ref
DENOVO Stainless Steel Crowns DENOVO DENTAL	Iron—69.22% Chromium—18.12% Nickel—9.58% Manganese—1.59% Silicon—0.52% Molybdenum—0.39% Copper—0.35% Cobalt—0.18% Carbon—0.05% Phosphorus—Trace Amounts Titanium—Trace Amounts Sulphur—Trace Amounts Aluminum—Trace Amounts Oxygen—Trace Amounts	Iron—30,000 mg/kg Oral—Rat Chromium—>9000 mg/kg Oral—Rat Nickel—n/a Manganese—9000 mg/kg Oral—Rat Silicon—n/a Molybdenum—n/a Copper—3160 mg/kg Cobalt—n/a Carbon—6171 mg/kg Oral—Rat Boric Acid—n/a Petroleum distillates—n/a Potassium Fluoborate—n/a Potassium Fluoride—n/a	[29]
Hu-Friedy PEDO CROWNS Hu-Friedy	Carbon—0.03% Sulphur—0.03% Silicon—0.75% Molybdenum—2.00% Phosphorus—0.045% Copper—0.22% Molybdenum—n/a Nickel—8.0–12% Chromium—18–20% Cobalt—n/a Iron—69.00%	n/a	[30]
Primary Stainless Steel Crowns 3M ESPE	Stainless steel 12597-68-1—100% Iron—65–74% Chromium—17–19% Nickel—9–13%	n/a calculated acute toxicity estimate >5000 mg/kg	[31] [32]
Unitek Primary SSC 3M ESPE	Stainless steel 12597-68-1—100% Iron—65–74% Chromium—17–19% Nickel—9–13%	n/a calculated acute toxicity estimate >5000 mg/kg	[31] [32]
KTR DentalCrown KTR	Iron—70–90% Chromium—15–35% Nickel—5–10% Manganese—2.5% Silicon—2.5% Copper—2.5%	n/a	[33]
Acero Stainless Steel Crown Acero Crowns	n/a	n/a	

**Table 2 materials-15-02081-t002:** The composition of zirconia crowns.

Brand	Composition	Toxicity	Ref
NuSmile Zr NuSmile Pediatric Crowns	Zirconium oxide—88–96% Yttrium oxide—4–6% Hafnium oxide—5% Organic Binder—2–5% Pigment—1–4%	n/a	[74]
Ez-Pedo Ez-Pedo	Zirconium oxide > 85% Hafnium oxide < 5% Yttrium oxide < 6% Organic binder < 5%	Acute oral toxicity: LD50 > 5000 mg/kg (rat) Data for zirconium oxide. Acute dermal toxicity: n/a Acute inhalation toxicity: LC50 > 4.3 mg/L (4 h exposure, rat) Data for zirconium oxide	[75]
Kinder Krowns Zirconia Kinder Krowns	Zirconium dioxide—70–100% Aluminum oxide—0–1% Yttrium oxide—1–5% Iron hydroxideoxid—0–3% Mixture of glycols < 1% Sodium, potassium, boron, and aluminum silicate glass < 1%	n/a	[76]

**Table 3 materials-15-02081-t003:** Summary of crowns used in pediatric dentistry [61].

Material	Available Brands	Advantages	Disadvantages
Stainless Steel	Hu-Friedy PEDO CROWNS Hu-Friedy Primary Stainless Steel Crowns 3M ESPE Unitek Primary SSC 3M ESPE Acero Stainless Steel Crowns Acero Crowns DENOVO SSC DENOVO DENTAL	1. Minimal tooth reduction required 2. High strength reliability 3. Good flexibility 4. Easy to contour and crimp 5. To improve the aesthetics, they can be used in open-faced crown technique	1. Low aesthetics 2. Possible nickel allergy and sensitivity
Pre-veneered Stainless Steel	NuSmile Signature NuSmile Pediatric Crowns Cheng Crowns Cheng Crowns Flex Crowns Success Essential Kinder Krowns Next Generation Kinder Krowns	1. High aesthetics 2. Metal edges can be crimped 3. Some brands provide personal customization	1. Pre-veneered material can crack from either crimping or wear-off during usage 2. Requires deeper preparation 3. Possible nickel allergy and sensitivity
Pre-veneered Aluminum	Pedo Pearls Java Crowns	1. High aesthetics 2. Minimal tooth reduction required 3. The tooth-colored coating is flexible which enables contouring and crimping	1. Lower strength 2. May offer decreased longevity 3. The tooth-colored coating is very thin and can wear off during use
Zirconia	Ez-Pedo Ez-Pedo NuSmile Zr NuSmile Pediatric Crowns Kinder Krowns Zirconia	1. Highest strength of all pediatric crowns 2. High aesthetic	1. Expensive 2. Cannot be crimped 3. Require deep preparation 4. Need to have good isolation for effective bonding
Polymer 1. Acrylic 2. Polycarbonate 3. Strip crowns	PedoNatural Crown Strip Crowns Forms 3M ESPE Pedo Jacket Crowns Success Essentials Pediatric Strip Crowns Success Essentials DirectCrown DirectCrown Products	1. Minimal tooth reduction required 2. Some brands offer crowns flexible enough to crimp	1. Due to decreased strength, their suitability for use in posterior location is questionable 2. Some polymerized polymers will not bond to placed resin

## Data Availability

Not applicable.

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
